# Learning to Reason and Memorize with Self-Notes

**Jack Lanchantin**[*]
Meta AI

**Shubham Toshniwal**[*]
NVIDIA

**Jason Weston**
Meta AI

**Arthur Szlam**
Meta AI

**Sainbayar Sukhbaatar**
Meta AI

## Abstract

Large language models have been shown to struggle with multi-step reasoning, and do not retain previous reasoning steps for future use. We propose a simple method for solving both of these problems by allowing the model to take *Self-Notes*. Unlike recent chain-of-thought or scratchpad approaches, the model can deviate from the input context at any time to explicitly think and write down its thoughts. This allows the model to perform reasoning on the fly as it reads the context and even integrate previous reasoning steps, thus enhancing its memory with useful information and enabling multi-step reasoning. Experiments across a wide variety of tasks demonstrate that our method can outperform chain-of-thought and scratchpad methods by taking Self-Notes that interleave the input text.

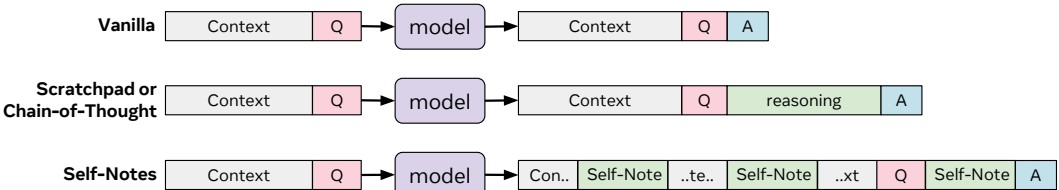

Figure 1: **[top] Vanilla** language models directly generate the answer (A) given the context and the question (Q). **[middle] Scratchpad and Chain-of-Thought** methods encourage the model to generate reasoning tokens before answering the question, but only after it has read the entire context. **[bottom] Self-Notes (ours)** allows the model to generate multiple internal reasoning notes that interleave the input context and question.

## 1 Introduction

Transformers [1] and similar variants have reshaped the field of machine learning with impressive results on sequence-based tasks [2]. Notably, large language models (LMs) such as GPT-3 [2] use transformers and are capable of solving various complex natural language tasks such as question answering. However, it's increasingly evident that there are still limitations to these models. Namely, transformers are limited in their ability to perform multi-step computations or store intermediate results due to the lack of an explicit internal dialogue or scratchpad [3, 4, 5].

When an LM is used for question answering (QA), it is typically fed a context prompt containing factual information along with a question, and then the model generates the answer directly, as shown in Fig. 1 (top). However, this autoregressive "one-step" approach struggles with multi-step reasoning

---

[*]Equal Contribution. Correspondence to sainbar@meta.com

37th Conference on Neural Information Processing Systems (NeurIPS 2023).

tasks [6, 7, 8, 4]. We argue that this arises from the fact that vanilla LMs have a fixed computation budget for each token, and do not have the option to "think" more depending on the current context.

Recently, Chain-of-Thought and Scratchpad methods [3, 4, 9, 10] encourage the model to generate reasoning tokens or explain their answer one step at a time, leading to improved reasoning capabilities. These extra tokens are added before answering the question, but *after* it has read the full context and question, therefore postponing all "thoughts" until the end, as illustrated in Fig. 1 (middle).

In addition to the "one-step" problem, transformers lack a recurrent memory for state-tracking and solving highly nonlinear tasks [11], something that recurrent predecessor models such as the LSTM [12] are well equipped for. Modifications to the feed-forward transformer architecture that use a recurrent mechanism improve state-tracking results [11, 13, 14], but still use a fixed computation amount for a given prompt.

In this paper, we propose an approach that simultaneously makes the challenges in multi-step reasoning and state-tracking memory more tractable. Our method, "*Self-Notes*", allows the LM to deviate from the context prompts at any time to generate explicit reasoning tokens. Unlike a scratchpad or chain-of-thought, the model can interleave generated tokens with the input context and question, as demonstrated in Fig. 1 (bottom). Such Self-Notes can act as both explicit intermediate reasoning steps and working memory for state-tracking. Overall, we see the ability of a model to think as it reads, storing those thoughts for later use, as an important aspect of an intelligent agent, similar to how humans read [15]. Consider answering a question after reading the contents of a book. One would not want to re-read and re-reason over the whole book again to answer a second question. This issue is not addressed by either vanilla language models or chain-of-thought approaches.

To give an example, when reading a text, at any time there may be a reasoning step that requires combining two facts. The resulting inference can be written into a Self-Note and used for future reasoning, thus acting as an intermediate reasoning step. For example, given "`Alice has the box`" and "`Alice is at the park`" one can infer "`The box is at the park`" and write it to a Self-Note, which can be further combined with a future statement "`The key is in the box`" to conclude that "`The key is at the park`". Additionally, the Self-Note can act as a form of working memory because the model can write the latest state of an entity as new tokens while it reads the context. For example, in a programming environment, assume `x=5` initially, and then x gets incremented by `1`, the model can write the new state `x=6` and add it to its working memory.

We test our method on seven text datasets designed to evaluate multi-step reasoning and state-tracking: a proposed synthetic Toy-Story task, two synthetic program evaluation tasks [11, 16], two real-world chess game tasks [17], and two math word problem tasks previously used to test chain-of-thought prompting, MultiArith and GSM8K [18, 19]. Across these tasks, we consider four different paradigms of learning Self-Notes: supervised, semi-supervised, unsupervised, and few-shot prompted. Across all tasks and learning paradigms, our method outperforms both an LM which does not do any explicit reasoning, as well as chain-of-thought or scratchpad baselines.

## 2   Method

Let us consider an autoregressive transformer model $\mathcal{M}$ that predicts the next token in a sequence,

$$x_{t+1} = \mathcal{M}(x_1, ..., x_t).$$

Such a model, $\mathcal{M}$ is the foundation of many tasks like language modeling and question answering. In such tasks, the model is given a context $C = \{x_1, ..., x_t\}$ and potentially a question $Q$ as input and asked to generate $A$ which is the sequence of next words or an answer to a question.

Our Self-Notes method expands the capability of $\mathcal{M}$ by allowing it to enrich context $C$ with "note tokens" $n_i$ before producing the final output $A$. Note tokens share the same vocabulary as input tokens, but they are generated by the model itself. Self-Notes generated in this way can interleave with the context tokens and therefore can be used for writing down a newly inferred fact or tracking variable values.

At inference time, while processing input tokens $x_t \in C$ one by one, the model can start taking a note by generating a token that belongs to a predefined set of start tokens $N_{sta}$. In other words, at any point in the input text, if the next most probable token is in $N_{sta}$, then the model can autoregressively generate itself a note. A note ends when the model generates an end token $n_i \in N_{end}$, or after a

Table 1: Input-Output pairs for the Vanilla, Scratchpad, and Self-Notes method on Toy-Story and Algorithmic tasks. The input consists of the `input context` and the `question`, the `answer` is to be generated by the model. The highlighted reasoning steps are not given at test time and is to be generated by the model.

| Task | Vanilla | Scratchpad | Self-Notes |
|------|---------|-----------|-----------|
| Toy-Story | Mary has the ball.
The ball is inside the box.
The key is inside the box.
Q: Who has the key?
Mary has the key. | Mary has the ball.
The ball is inside the box.
The key is inside the box.
Q: Who has the key?
[Q: Who has the box?
Mary has the box.
Q: Who has the key?
Mary has the key.]
Mary has the key. | Mary has the ball.
The ball is inside the box.
SQ: Who has the box?
Mary has the box.
The key is inside the box.
SQ: Who has the key?
Mary has the key.
Q: Who has the key?
Mary has the key. |
| Algorithmic | e = 3 ; e ++ ;
i = 3 ; if i < e :  e ++ ;
print e e = 5 ; | e = 3 ; e ++ ;
i = 3 ; if i < e :  e ++ ;
print e
[e = 3 ; e ++ ;
print e e = 4 ;
i = 3 ; if i < e :  e ++ ;
print e e = 5 ;] e = 5 ; | e = 3 ; e ++ ;
print e e = 4;
i = 3 ; if i < e :  e ++ ;
print e e = 5;
print e  e = 5 ; |

fixed number of tokens are generated. Once the note ends, the generated note tokens are appended to the context where the start token was generated, and the model continues to process the rest of the input tokens. For example, a context $C = \{x_1, x_2, x_3, x_4\}$ can be enriched to become $\{x_1, x_2, n_1, n_2, n_3, x_3, x_4\}$ if the start token is generated after $x_2$:

$$n_1 = \mathcal{M}(x_1, x_2) \in N_{\text{sta}}, \quad n_2 = \mathcal{M}(x_1, x_2, n_1) \notin N_{\text{end}}, \quad n_3 = \mathcal{M}(x_1, x_2, n_1, n_2,) \in N_{\text{end}}.$$

By repeating this mechanism, the context $C$ can be enriched with multiple notes at different locations. An overview of our method is shown in Figure 1 (bottom). A detailed example of how the model takes Self-Notes at inference is shown in Figure 5 (bottom).

The model can use notes as a form of working memory by writing information that might be useful in the future. It can also use a note as an intermediate reasoning step by inferring new facts as it reads. In particular, it can ask a question and answer it within it. This is useful in multi-step reasoning where a final question requires answering multiple sub-questions. Unlike implicit reasoning occurring internally within $\mathcal{M}$, Self-Notes are fed back to the model, making it available to future reasoning steps. This feedback loop also allows the model to overcome the limitation of transformers as a feedforward network [11], making it possible to do state-tracking. We explain four different paradigms to encourage the model to write Self-Notes as follows.

**Supervised Self-Notes.** One way to train $\mathcal{M}$ to generate useful notes is to use supervised learning on data that is enriched with "ground-truth" Self-Notes interspaced within the context. This training procedure is simple as we just have to train $\mathcal{M}$ on this enriched data using the standard LM training loss. After training, we can use $\mathcal{M}$ to generate Self-Notes, so we can apply it to test data that does not contain any Self-Notes or reasoning labels. $\mathcal{M}$ can generate a Self-Note at test time by predicting the next token in the context to be from $N_{\text{sta}}$.

**Semi-supervised Self-Notes.** We also consider a semi-supervised setting where only a subset of the training samples have ground truth Self-Notes. In this case, we prepend a special token $s$ to training samples without Self-Notes and train all samples with the standard LM loss: $C = \{s, x_1, ..., x_t\}$. As a result, the model is conditioned to generate Self-Notes during test time because the test context does not contain the special token $s$ prefix. This signals to the model that it should do extra reasoning and generate Self-Notes.

**Unsupervised Self-Notes.** We introduce a method for utilizing Self-Notes when no ground truth note is available for training.[2] This method relies on the fact that when the model is trained using the LM loss on all tokens in a QA task, it learns to not only generate answers but also questions. We leverage this property by letting the model generate its own questions and insert their answers as Self-Notes (i.e., interleaved throughout the context) during test time.

---

[2] We call this method "unsupervised" or "unlabeled" Self-Notes because there is no supervision or labeling on the notes, but there is full supervision on the output answers.

If we train the model to predict the final question and answer with varying length samples, the model will learn to generate a question after any number of statements. At the same time, we allow the model to write a Self-Note after each intermediate statement. Assuming the model has learned how to answer the shorter samples, it is likely to write the correct value in the intermediate locations. It can then leverage that information on the longer samples. If the relevant intermediate questions are asked and answered, this will make it easier to answer the final question.

We consider two potential problems with approach. The first is that as the context is enriched by Self-Notes, it can become longer than what the model has seen during training, or it can contain new tokens that it didn't see in the context during training. A simple solution is to finetune the model on the Self-Notes enriched samples during training. The training procedure therefore has two simultaneous objectives: learn how to write Self-Note QA pairs after any number of context tokens, and leverage the written Self-Note answers for the final question. The second problem is that the model might not ask enough questions because training samples contain only one final question. We solve this by simply amplifying the probability of generating a Self-Note start token (from $N_{\text{sta}}$), by a multiplicative "boosting" constant $B > 1$. Furthermore, we can sample multiple versions of Self-Notes per sample and select the enrichment that leads to the most confident answer.

**Self-Note Prompting of Large Language Models.** Lastly, we consider prompting a pretrained large language model (LLM) with Self-Notes. In this setting, we show the model few demonstrations of how to write Self-Notes in the prompt. During inference, we allow the LLM to insert a Self-Note every time the start token is predicted as the next token within the current context. This is similar to chain-of-thought prompting [4], except we replace the chain-of-thought reasoning that comes after the question with Self-Notes that can be anywhere in the context. Once all the Self-Notes are generated, we let the LLM generate its final answer.

# 3 Experiments

We compare against two baseline methods: a vanilla transformer language model, and a transformer language model trained to generate a chain-of-thought "scratchpad". For all experiments except few-shot prompting, we use the following LMs. The *Vanilla* baseline is the pretrained GPT-2 base model [20] from Hugging Face [21] fine-tuned to predict answer tokens given only the context and question. For the *Scratchpad* (i.e. Chain-of-thought) baseline, we fine-tune the same GPT-2 model to write a scratchpad of reasoning steps after it has seen the context and question, similar to Nye et al. [3]. For the proposed *Self-Notes* model, we fine-tune GPT-2 to take Self-Notes. During testing, no ground-truth scratchpad or Self-Notes are provided, but both Scratchpad and Self-Notes models are allowed to generate tokens in addition to the answer.

## 3.1 Tasks

In this section, we explain each task we test our models on. Table 1 shows a sample for two task with different methods: Vanilla, Scratchpad, and Self-Notes. For each task, we evaluate on both an in-distribution and out-of-distribution (OOD) test set. More detailed descriptions of each dataset and a summary of dataset statistics is provided in the Appendix.

**Toy-Story.** Inspired by the bAbI tasks [22], we introduce a new synthetic QA task for testing the ability of language models to do multi-step reasoning. The task is to answer a question after reading a short story that consists of multiple sentences. The challenge in this dataset is that by applying pragmatic principles, unseen relations can be inferred from observed relations. For example, given the text "`Alice is at the park. Bob is with Alice.`", we can infer that "`Bob is at the park.`". Furthermore, a newly inferred relation can lead to inference of another unseen relation. In the previous example, if the next sentence is "`Bob has the key.`", then we can infer that "`The key is at the park`" using our previous inference about Bob's location. This recursive inference in Toy-Story makes it possible to create questions that require multi-step reasoning.

We use Self-Notes to infer all implied relations. Within a Self-Note, the model can ask and answer a question, e.g., "`SQ: Where is Bob? Bob is at the park.`". The Scratchpad method should infer the same relations, but it will be forced to do it after the question is asked, requiring backward-reasoning. To test generalization, we train the model on 10k 1-hop and 2-hop queries, and test on 3-hop and 4-hop queries.

Table 2: Test Accuracy (in %) for the reasoning and state-tracking tasks in the supervised setup. "*" indicates out-of-distribution harder test settings.

| Task | Test Set | Vanilla | Scratchpad | Self-Notes |
|---|---|---|---|---|
| **Toy-Story** | **1/2-hop** | 92.4 ±0.7 | **99.6** ±0.1 | **99.8** ±0.1 |
| | **3-hop*** | 57.0 ±0.3 | 96.4 ±0.9 | **98.5** ±0.3 |
| | **4-hop*** | 37.4 ±0.8 | 94.2 ±2.0 | **97.8** ±0.4 |
| **Algorithmic** | **2-100** | 44.6 ±1.0 | 72.2 ±5.7 | **95.5** ±0.2 |
| | **101-200*** | 24.4 ±2.1 | 11.6 ±2.0 | **85.0** ±0.6 |
| **Boolean Variable** | **3-8** | 99.7 ±0.1 | **100.0** ±0.0 | **100.0** ±0.0 |
| | **9-19*** | 71.3 ±0.8 | 73.7 ±2.4 | **75.2** ±2.1 |
| **Chess Piecetype** | **≤80** | 98.5 ±0.4 | 98.5 ±0.3 | **98.8** ±0.2 |
| | **≥81*** | 82.9 ±2.3 | **94.7** ±0.7 | **94.8** ±0.7 |
| **Chess Move** | **≤80** | 49.0 ±0.4 | 37.0 ±0.8 | **50.8** ±1.1 |
| | **≥81*** | 39.8 ±0.2 | 29.9 ±0.8 | **41.8** ±0.9 |

**Algorithmic.** To evaluate the ability of tracking the state or value of an entity over multiple steps, we adopt the Algorithmic task from [11], where the context is the sequence of statements, e.g. "`x = 1 ; x++ ; y = 3 ; if x > 2:   y++ ;`", and the final question is to print the last value of one of the variables, e.g. "`print x`". While the original task has separate input and label tokens, we unify them into a single sequence to fit the language modeling task.

For the Self-Notes model, the notes are print statements specifying the intermediate value of a certain variable as they are modified. For example, if the previous statement was "`x++ ;`", the Self-Note would be to print the value of `x`: "`print x x = 1 ;`". The Scratchpad method generates the identical print statements, but it also has to copy all of the original statements to the scratchpad and figure out where to insert the prints, thus introducing an additional "alignment" complexity in comparison to the Self-Notes method. We train the models on 2 to 100 statements in each sample. We test on 2-100 (in-distribution) and 101-200 (OOD) statements.

**Boolean Variable.** In this task, the context consists of a valid Python program where each statement contains a *boolean* variable assignment operation. The question is to print the final value of an arbitrary variable (True or False). The main difference to the Algorithmic task is that the statements are constrained to boolean logic operations. We use the "chain-like" split from Anil et al. [16], which consists only of operations that compose the values of already defined variables. This results in long chains of dependencies between values of the variable. Similar to the Algorithmic task, Self-Notes prints the value of the variable that was modified in the previous statement, e.g. "`print x True;`". Following Anil et al. [16], we train on 3-8 statements and test on 3-8 and 9-19 statements.

**Chess Piecetype.** The goal of this task is to track the state of chess pieces over a sequence of moves in a real chess game [17]. Chess games written in UCI notation consist of a sequence of (start position, end position) pairs, e.g. `c2 c4`, which denotes a move of the piece from `c2` to `c4`. The piecetypes of the moves are never given explicitly, but since each game begins with pieces in the same positions, the piecetypes can be implicitly tracked given the moves. Given a long sequence of moves, e.g. "`c2 c4 e7 e5 g2 g3 b8 c6 f1 g2 g8 f6 b1 c3 f8 b4 c6`", the objective is to predict the piece at the last position mentioned ("`c6`").

For our proposed method, we consider the Self-Notes to be the piecetypes. A Self-Note is inserted after the start position of each move to explicitly remind the model which piecetype is at that position. So the previous example would look like: "`c2 P c4 e7 P e5 g2 P g3 b8 N c6 f1 B g2 g8 N f6 b1 N c3 f8 B b4 c6`", and it is therefore much easier with Self-Notes to predict the piecetype at "`c6`", since we know that the last piece moved to "`c6`" was a knight during the move "`b8 N c6`". To test length generalization, we consider more moves during testing than seen during training.

**Chess Move.** This task is to predict the end position of the current move given the start position [17]. For example, given the sequence of moves "`c2 c4 e7 e5 g2 g3 b8 c6 f1 g2 g8 f6 b1 c3 f8 b4 c6`", the answer is the ground truth final position made in the game: "`e5`". This task is harder than the Chess Piecetype task as the model needs to learn, state tracking, chess rules, and chess strategy in order to predict the most likely move. The Self-Notes are the same as chess piece, where the model is trained to generate the piece at each starting square as it makes a move, e.g. "`c2 P c4`".

**Math Word Problems.** For the few-shot prompting, we consider two real-world math tasks that require solving multi-step elementary-level arithmetic word problems: MultiArith [18] and GSM8K [19]. For example, the following example in the MultiArith format requires language understanding, as well as tracking the value of the entities: "`Alice had 4 cupcakes and 2 cookies. Alice then ate 1 cupcake. After Alice gives 1 cookie to Bob, how many treats does she have left?`". In the few-shot prompt, we wrote useful intermediate reasoning and calculations in Self-Notes surrounded by parenthesis. For example the Self-Notes for the previous sample should look like: "`Alice had 4 cupcakes and 2 cookies.` `(4 + 2 = 6 total)` `Alice then ate 1 cupcake.` `(6 - 1 = 5 left)` `After Alice gives 1 cookie to Bob, how many treats does she have left?` `(5 - 1 = 4 left)`". For MultiArith, we use the few-shot training samples from [4]. GSM8K contains more challenging problems that requires more diverse reasoning. We selected 5 random samples in the training set to use as the few-shot demonstrations in the prompt. For Chain-of-Thought, we used the original solutions (minus the calculation annotations written inside "« »") given in the dataset. For Self-Notes, we manually annotated the few-shot examples with Self-Notes (shown in the Appendix).

## 4 Results

### 4.1 Supervised Self-Notes

Table 2 shows the results for the five tasks described in Section 3.1.

**Toy-story.** For both the 3-hop and 4-hop settings, we see that the Self-Notes model substantially outperforms the Vanilla model which has to perform multi-step reasoning in "one-step". We observe a slight improvement of the Self-Notes model over the Scratchpad model. We reason that the drop in Scratchpad's performance has to do with the model having to postpone until after processing the entire input context, which increases the distance between the input context and the reasoning. In comparison, the Self-Notes model writes reasoning tokens on the fly as the relevant facts are stated. We note that for this task, the full context fits into the GPT-2 context window.

**Algorithmic.** We observe that the Vanilla GPT-2 model struggles to track the state of the variables over many statements, and significantly worsens for OOD sequence lengths. Self-Notes, which allows the model to generate intermediate print statements, achieves high accuracy on both the in-distribution and OOD statement splits. Scratchpad fails at most examples since the context length exceeds the maximum length of GPT-2 (1024 tokens). This leads to a worse performance than the Vanilla model because it tries to write the scratchpad which involves copying the original context, but then runs out of room and can't answer the question. These results show a significant advantage of our method: as long as the model takes a Self-Note about a variable, it will keep it in the memory by pushing its value to the most recent context. The Scratchpad method has to copy the entire context in its scratchpad, often going past the maximum context length, resulting in poor accuracy.

**Boolean Variable.** Unlike the Algorithmic task, none of the models run out of context length for this task since there are fewer statements. Therefore, Scratchpad is able to perform similarly to Self-Notes. However, we still see a small increase in performance with Self-Notes, likely due to copy alignment errors in the Scratchpad. Both the models improve over Vanilla.

**Chess Piecetype and Chess Move.** The chess tasks primarily measure the ability of a model to track the identity and state of variables over a sequence of changes. In the Chess Piecetype task, both the Self-Notes and Scratchpad models outperform the Vanilla model. As with other tasks, this confirms that Vanilla transformers are improved with extra tokens in order to accurately track the state of a set of variables, particularly when the test-time sequence lengths vary from the training length. For Chess Piecetype, Self-Notes is not significantly better than Scratchpad. This is a fairly simple task for Scratchpad since it simply requires copying the piece at each move, assuming it knows where the pieces start. This is different from the Algorithmic and Boolean Variable tasks which not only need to copy the variable, but also increment, decrement, or negate it.

In the Chess Move task, Self-Notes is slightly better than Vanilla, but Scratchpad is significantly worse than both. In this task, the Self-Notes and Scratchpad "note" tokens (pieces) are not the same as the final question (move board position). We hypothesize that Scratchpad cannot learn to simultaneously copy the identity of pieces *and* predict the chess move.

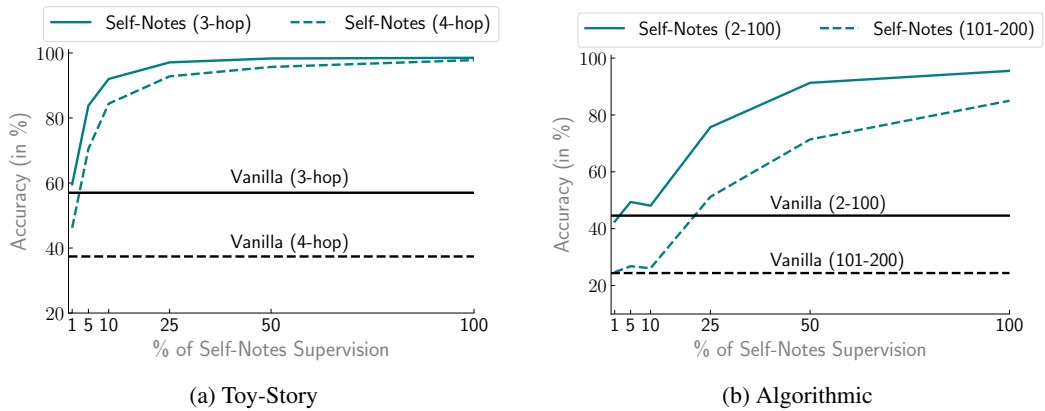

(a) Toy-Story

(b) Algorithmic

Figure 2: Performance of the Semi-supervised Self-Notes method with varying amounts of Self-Notes supervision on the Toy-Story and Algorithmic tasks.

Table 3: Unsupervised Self-Notes: Toy-Story task accuracy (%)

| Method | 3-hop | 4-hop |
|---|---|---|
| Vanilla | 79.4 | 57.9 |
| + Self-Notes (unsupervised) | 79.7 | 61.8 |
| + Boost Questions | 82.4 | 68.2 |
| + Multi-sample | 91.3 | 79.1 |
| + Finetune | 94.2 | 85.8 |

## 4.2 Semi-supervised Self-Notes

Figure 2 shows the performance of the Self-Notes method with varying amounts of Self-Note supervision for the Toy-Story and the Algorithmic tasks. That is, we randomly sample some percentage of the training samples that get Self-Note supervision. For Toy-Story, we find that even Self-Note supervision using as little as 1% of the training set (100 samples), leads to performance gains over the Vanilla model, and the performance starts to saturate around 25% supervision. On the other hand, for the Algorithmic task, we observe gains with Self-Note supervision starting at around 5% supervision, and the performance steadily improves with more Self-Note supervision.

## 4.3 Unsupervised Self-Notes

In our third set of experiments, we apply Self-Notes to a 1-variable Algorithmic task and Toy-Story task when we have no ground-truth Self-Notes to train on.

First, we conduct experiments in the unsupervised setting for Algorithmic task. We first test on a 1-variable version of the dataset (each sample contains only 1 variable). We train on datasets that contain varying length samples (i.e. varying numbers of algorithmic statements per sample), so the model will generate intermediate Self-Notes on its own in a QA form. In this task, we allow the model to generate Self-Notes, and then conditioning on the previous note to predict the next Self-Note and final answer during training, departing from the standard parallel training procedure. The model therefore has to do two simultaneous tasks during training, write the correct Self-Notes, and predict the final answer given the written Self-Notes. Since we only use 1-variable samples, it makes it straightforward to learn which Self-Note questions to write (it will always be `print x`, where x is the variable in that sample. We can see from Figure 3, that around 10k samples, the unsupervised Self-Notes method starts to learn how to write and leverage Self-Notes that improve accuracy over the Vanilla method. With 20k samples, the unsupervised Self-Notes method achieves near 100% accuracy, with a significant increase over the Vanilla model.

The second task we consider for unsupervised Self-Notes is Toy-Story. Here, the training data has 100k samples with 1 and 2 hop questions, but contains no Self-Notes. This task is more difficult since there are many more variables (people, objects, locations) and model needs to ask the right questions in Self-Notes.

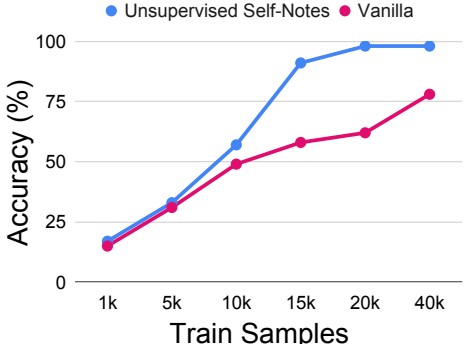
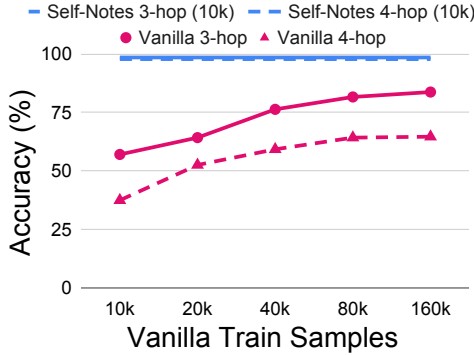

Figure 3: Unsupervised Self-Notes vs Vanilla sample comparison on 1-variable Algorithmic.

Figure 4: 10k Supervised Self-Notes samples vs Vanilla sample comparison on Toy-story.

We first train a Vanilla model to generate the final question and answer, with test accuracy shown at the top of Table 3.[3] Next, we test the vanilla model with Self-Notes by allowing it to generate QAs as notes. Here, we only add the answer parts to the context because the model has never seen a question in the context during training. Additionally, because the model is trained on 1-hop questions, it often asks a question whose answer is already in the context. We ignore such duplicate answers and move on to the next context sentence.

Simply adding Self-Notes to the Vanilla model during testing does not improve the performance much because the model is not trained to ask the right questions. We therefore encourage the model to ask more questions by boosting the probability of the question start token "Q:". Boosting Self-Notes by $B$=5 does improve the performance over the Vanilla model. Furthermore, generating multiple different Self-Notes by sampling questions and selecting the most confident one also helps. This is likely because when the right question is asked and answered, the model becomes more confident in its final answer. Finally, finetuning the model on a Self-Note version of the original training data improves the performance, as it adapts to longer stories. In summary, we see a significant increase in accuracy over the Vanilla results by allowing the model to generate Self-Notes and finetuning the model on the generations.

### 4.4 Few-shot Prompted Self-Notes

For the MultiArith task, we test a few-shot prompted GPT-J model [23]. For the Vanilla and Scratchpad baselines, we use the same few-shot prompt demonstrations from [4]. For Self-Notes, we use the same few-shot examples, but additionally annotated with Self-Notes that we wrote (provided in the Appendix). In Table 4, we compare Self-Notes to the Vanilla and Chain-of-Thought (CoT) methods for similar sized models (6.7B-8B). We find that the Self-Notes method on the 6B GPT-J model outperforms all of the Vanilla and CoT results. This result highlights two key findings. First, Self-Notes can write pertinent, yet mathematically accurate notes on arithmetic word problems. Second, we can effectively use Self-Notes on pretrained LLMs using few-shot prompting.

For the more challenging GSM8K task, we employed a GPT-3 model with 175B parameters. As shown in Table 5, Self-Notes outperforms CoT, but by a small margin. Since we prompted the model using the public API and had no access to token output probabilities over the prompt itself, we limit Self-Notes positions to immediately after ".", ",", or "?" to reduce the API calls. See [4] for more comprehensive comparison of CoT against Vanilla prompting (note that they used a more powerful version of GPT-3 than us).

Lastly, we run few-shot prompting experiments using the state-of-the-art open model, Llama 2 (70B) [24]. We compare Vanilla, Chain-of-Thought, and Self-Notes across three datasets: Algorithmic, MultiArith, and GSM8K. For GSM8K and MultiArith, we use the 8-shot arithmetic reasoning prompt from [4]. The CoT and Vanilla prompts are the same as from [4] and the Self-Notes prompt adapts the CoT prompt by putting some of the reasoning steps inside the context. For Algorithmic, we use a 5-shot prompt of randomly selected examples from our training set, ranging from 6-33 statements.

---

[3]Here we see better Vanilla performance than Table 2 because the training data is larger.

Table 4: GPT-3, LaMDA, PaLM, and GPT-J few-shot prompting accuracy (%) on MultiArith.

| Model | Size | Vanilla | CoT | Self-Notes |
|-------|------|---------|-----|-----------|
| **GPT-3** [4] | 6.7B | 4.5 | 2.8 | - |
| **LaMDA** [4] | 8B | 5.8 | 1.5 | - |
| **PaLM** [4] | 8B | 4.2 | 15.8 | - |
| **GPT-J** | 6B | 6.0 | 13.5 | **21.0** |

Table 5: GPT-3 few-shot prompting accuracy (%) on GSM8K.

| Model | Size | CoT | Self-Notes |
|-------|------|-----|-----------|
| **GPT-3** | 175B | 11.4 | **13.4** |

Table 6: Llama 2 (70B) few-shot prompting accuracy (%).

| Dataset | Vanilla | CoT | Self-Notes |
|---------|---------|-----|-----------|
| Algorithmic | 23.3 | 37.4 | **40.3** |
| MultiArith | 33.3 | 94.5 | **96.2** |
| GSM8K | 16.5 | 55.9 | **59.9** |

We test on the 2-100 statement test set from our paper. Results are shown in Table 6. In summary, we find that Self-Notes outperforms Chain-of-Thought and the Vanilla baseline on all three tasks.

### 4.5 Ablation: Labeled training set size comparison

Each Self-Notes training sample in Toy-Story has intermediate questions and answers, therefore increases the total number of QA pairs in the training set. For example if the final question and answer is "`Where is the ball? The ball is in the park`", but there was a Self-Note in the middle of the context labeled "`Who has the ball? Alice has the ball`", then that sample has two QA pairs. We therefore also run a comparison of the total number of labelled QA pairs between Self-Notes and Vanilla. Specifically, the 10k Self-Notes training data for Toy-Story has 10k total samples, and therefore 10k final QA pairs. However, it also includes roughly 70k Self-Note QA pairs which means that the total amount QA pairs is around 80k. Figure 4 shows the effect of increasing the training size for the Vanilla baseline compared to a fixed set of 10k Self-Notes training samples (80k labeled QA pairs) in the Toy-Story task. The Self-Notes model with 10k samples still vastly outperforms the Vanilla model with a 1500% increase in training samples (and roughly a 100% increase in the amount of QA pairs to that of Self-Notes). See Appendix for more ablation experiments, including understanding the difference between content and compute in Self-Notes by using "dummy" tokens (Appendix subsection A.3).

## 5 Related Work

**Implicit Reasoning.** bAbI [22] was a set of synthetic tasks for testing different reasoning capabilities [25] and showed the advantage of attention-based models over recurrent neural networks [26, 27]. More recently, attention-based transformers [1] have become the foundation of language-based reasoning [28]. However, the feedforward nature of transformers makes it unsuitable for state-tracking [11] and several recurrent versions have been proposed [29, 13, 14]. Further, transformer-based large LMs are shown to struggle at multi-step reasoning [7].

**Explicit Rationales.** Use of rationales has been explored for interpretability [30], and for performing intermediate computations [31, 3, 4]. In particular, the Scratchpad method by Nye et al. [3] is closest to our proposed Self-Notes method which can be interpreted as an *online*-variant of Scratchpad. Use of rationales for reasoning and arithmetic tasks, referred to as "chain-of-thought", has been shown to be particularly beneficial for zero- and few-shot in-context learning with large language models [4, 32, 7]. Zelikman et al. [9] showed the possibility of bootstrapping from a small set of reasoning labels to a larger unlabeled dataset. Trivedi et al. [33] propose interleaving chain-of-thought reasoning with knowledge retrieval steps (both after context). Other works have generated "inner monologue" tokens as a form of intermediate reasoning after a prompt [10, 34]. However, as with Scratchpad, the "chain-of-thought" and "inner monologue" reasoning is done after reading the entire input context rather than while reading it as in Self-Notes. This emergent capability of chain-of-thought prompting is only possible in large ($> 6B$) models [4]. In concurrent work, Toolformer [35]

introduces a method to finetune LMs to make API calls to external tools in the middle of the context. We propose a more general method to augment the context without API calls.

**Length Extrapolation.** Length extrapolation, or generalization to longer instances during inference than those seen during training is an important property for an intelligent agent. In the context of transformer-based models, length extrapolation has been explored for language modeling [36], machine translation [37, 38], models trained on artificial datasets [39, 16], and a variety of other tasks. One of the reasons for the limited length generalization capability of transformer models is due to the input position being represented with learnable embeddings [38, 40].

**Adaptive Computation.** When humans read and write text, we often spend a different amount of time per sentence. Transformers are designed to process and generate each sentence using the same amount of computation regardless of the complexity. Several works have addressed the problem of fixed computation time [41, 42, 43], but require modifying the training procedure or model architecture. Self-Notes can be viewed as a form of adaptive computation because the model decides when to deviate from the context and "think". Unlike previous adaptive computation approaches, Self-Notes can easily be applied to existing LM architectures and training procedures.

**Editing of Generated Text.** Several works have introduced variants of the transformer architecture to allow insertions, deletions, and revisions to the generated text [44, 45, 46, 47, 48]. Contrary to these methods that revise post-context generations, Self-Notes revises the original prompt context by inserting tokens on the fly.

## 6    Conclusion

We proposed a general method that allows language models to explicitly reason and memorize in the form of taking Self-Notes. Unlike scratchpad and chain-of-thought methods that postpone reasoning until all input tokens are processed, our method is more general and can deviate from the input sequence at any time for writing a Self-Note. One advantage of interleaving reasoning with the context in this way is that the reasoning steps can be closer to their relevant context. Another advantage is that it can act as a recurrent memory as the Self-Note answers are fed back to the model. Both these advantages make the method generalize better than previous approaches, as shown in our experiments. We demonstrate this improvement with four different learning paradigms: supervised, semi-supervised, unsupervised, and few-shot prompted, using four different models of varying sizes. We note that scratchpad and few-shot chain-of-thought also require additional human annotations. So, while it is a drawback compared to vanilla training, it requires the same amount of additional annotation as scratchpad/CoT. The goal of this work is to experimentally validate the difference between in-context thoughts (Self-Notes) vs purely post-context thoughts (CoT/Scratchpad). Future work should explore two complementary directions aimed at reducing the amount of supervision: (1) using reinforcement learning to discover the optimal Self-Notes, and (2) whether future models can generate relevant Self-Notes out of the box. Our results suggest that large language models can generate meaningful Self-Notes via few-shot prompting. This excites us about the applications that Self-Notes will enable in the future.

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

# A Appendix

## A.1 Tasks

Here we explain the tasks in more detail than outlined in Section 3.1. A summary of dataset statistics is provided in Table 8.

**Toy-Story.** As we read a story, we are often required to infer things that are not explicitly mentioned [15]. For example, reading "Frodo went to Mount Doom. Sam accompanied him.", a reader can infer that "*Sam went to Mount Doom*". Making such inferences in an online manner as we're reading the story makes it easier to understand the rest of the story. Such forward reasoning is natural for understanding sequential stories like books or movies. It is also more fitting for dialog models as such a model needs to make inferences as conversation happens and respond accordingly. In contrast, backward reasoning starts with a question and tries to find the relevant facts from a given context to answer it, potentially leading to a more narrow understanding of context.

Toy-Story is a synthetic QA task that can be used to test the ability of language models to perform forward reasoning. Each sample consists of a short story followed by a question and its corresponding answer. Sentences in the story state a simple relation between people, items, and places such as "`Alice is at the park`" or "`The ball is in the bag`". There are 5 different types of relations. This dataset is inspired by the bAbI tasks [22], with greater controllability of required reasoning steps. Unlike bAbI, our dataset mixes different reasoning steps to create more "hops" in order to answer a question.

We call a question $k$-hop if it requires $k$ observations combined through $k$-1 reasoning steps (1-hop questions only require repeating of an observed fact). While a backward reasoning model needs to take $k$ reasoning steps to answer a $k$-hop question, a forward reasoning model will infer unseen relations as each sentence is processed. As a result, forward reasoning can uncover all relations by the end of the story and can answer any question with no additional reasoning.

Considering the relevance of forward-reasoning to the Toy-Story task, Self-Notes is therefore a natural fit. Self-Notes should explicitly infer all implied relations. For this dataset, the Self-Note start and end tokens are "`SQ:`" and "`.`", respectively. Following the start token, the model can ask and answer a question, e.g., "`SQ: Where is Bob? Bob is at the park.`".

If the model correctly learns to infer relations during training, then it can easily answer 3 and 4-hop queries by inferring the intermediate (2-hop) relations. Specifically, by writing a Self-Note, the model can turn a 3-hop query into two separate 2-hop queries.

A sample from the test set is shown in Table 9, where the Vanilla and Scratchpad models fail to predict the correct answer, but the Self-Notes model correctly writes the answer in its notes.

**Algorithmic.** While the Toy-Story task is designed for testing multi-step reasoning, it doesn't require tracking the state or value of an entity over multiple steps since it assumes the world is static. Algorithmic task from [11] requires printing the state, or value, of an *integer* variable given a sequence of algorithmic program statements such as increment, decrement, and conditionals. We unify the input and label tokens into a single sequence to fit the language modeling task. The start token is "`print`" and the end token is "`;`".

**Boolean Variable.** In this task, the context consists of a valid Python program where each statement contains a *boolean* variable assignment operation. The question is to print the final value of an arbitrary variable (True or False). So the Self-Notes look like: "`print x False ;`". Like Algorithmic, the start token is "`print`" and the end token is "`;`". The main difference to the Algorithmic task is that the statements are constrained to boolean logic operations. We show an example of the Self-Notes and Scratchpad samples for this task in Table 7.

**Chess Piecetype.** This is another state-tracking task designed to to track the state of chess pieces over a sequence of moves in a real chess game [17]. Chess games written in UCI notation consist of a sequence of (start position, end position) pairs, e.g. `c2 c4`, which denotes a move of the piece from `c2` to `c4`.[4] The piecetypes are never given in the samples, but can be implicitly tracked given the

---

[4]To ease tokenization, we split a move in UCI notation from `c2c4` to `c2 c4`. We add all the 64 board squares to the language model's vocabulary.

Table 7: Input-Output pairs for the Vanilla, Scratchpad, and Self-Notes method for the non-prompting tasks that were not listed in Table 1. The input consists of the `input context` and the `question`, the `answer` is to be generated by the model. The highlighted text for Scratchpad and Self-Notes is available only during training, and is to be generated by the model during inference.

| Task | Vanilla | Scratchpad | Self-Notes |
|------|---------|-----------|-----------|
| Boolean Variable | `w = False ; v = True ;`
`v = w xor v ;`
`w = v and v ;`
`print w True ;` | `w = False ; v = True ;`
`v = w xor v ;`
`w = v and v ;`
`print w`
`[w = False ; v = True ;`
`v = w xor v ;`
`print v True ;`
`w = v and v ;`
`print w True ;] True ;` | `w = False ; v = True ;`
`v = w xor v ;`
`print v True ;`
`w = v and v ;`
`print w True ;`
`print w True ;` |
| Chess Piecetype | `c2 c4 e7 e5 g2 g3 b8 c6 f1 g2`
`g8 f6 b1 c3 f8 b4 c3`
`PIECE N` | `c2 c4 e7 e5 g2 g3 b8 c6 f1 g2`
`g8 f6 b1 c3 f8 b4 c3`
`PIECE`
`[c2 P c4 e7 P e5 g2 P g3`
`b8 N c6 f1 B g2 g8 N f6 b1`
`N c3 f8 B b4 c3] N` | `c2 P c4 e7 P e5 g2 P g3`
`b8 N c6 f1 B g2 g8 N f6 b1`
`N c3 f8 B b4 c3`
`PIECE N` |
| Chess Move | `c2 c4 e7 e5 g2 g3 b8 c6 f1 g2`
`g8 f6 b1 c3 f8 b4 c3`
`MOVE c5` | `c2 c4 e7 e5 g2 g3 b8 c6 f1 g2`
`g8 f6 b1 c3 f8 b4 c3`
`MOVE`
`[c2 P c4 e7 P e5 g2 P g3`
`b8 N c6 f1 B g2 g8 N f6 b1`
`N c3 f8 B b4 c3] c5` | `c2 P c4 e7 P e5 g2 P g3`
`b8 N c6 f1 B g2 g8 N f6 b1`
`N c3 f8 B b4 c3`
`MOVE c5` |

moves. For example, since each game starts with a pawn (`P`) at board position `c2`, we know that given the move `c2 c4`, there is now a pawn at position `c4`. Given a long sequence of moves, e.g. "`c2 c4 e7 e5 g2 g3 b8 c6 f1 g2 g8 f6 b1 c3 f8 b4 c6`", the objective is to predict the piece at the last position mentioned ("`c6`"). For Self-Notes, the notes are the piecetypes. So the previous example would be written as "`c2 P c4 e7 P e5 g2 P g3 b8 N c6 f1 B g2 g8 N f6 b1 N c3 f8 B b4 c6`". For the chess tasks, the start token is any of the pieces and there is no end token (since the note is only 1 token). To differentiate the Self-Notes from the final question, which also asks for the piecetype, the final question is asked using the `PIECE` token. Samples for this task are shown inin Table 7.

We consider a different number of moves during training and testing to test length generalization. We train our models on 200k samples which include up to 80 moves. We evaluate on both up to 80 moves (in-distribution) as well as more than 80 moves (OOD).

**Chess Move.** This task is to predict the end position of the current move given the start position [17]. The Self-Notes are the same as Chess Piecetype, where the model is trained to generate the piece at each starting square as it makes a move. We report the exact match accuracy. We use the same train/valid/test split as the Chess Piecetype task. Table 7, shows an example of this task, which is similar to Chess Piecetypewith the exeption of the question token, `MOVE` as well as the final answer being a board position.

**MultiArith.** This is a real-world reasoning dataset was introduced to evaluate a model's ability to solve multi-step arithmetic word problems [18]. An example from the dataset is "`While on vacation, Megan took 15 pictures at the zoo and 18 at the museum. If she later deleted 31 of the pictures, how many pictures from her vacation did she still have?`". Here, the Self-Notes are intermediate summaries in the middle of the context. For example in the previous example, they would look like: "`While on vacation, Megan took 15 pictures at the zoo and 18 at the museum.` `(18 + 15 = 33 total)` `If she later deleted 31 of the pictures, how many pictures from her vacation did she still have?` `(33 - 31 = 2 left)`". Here, the start token is "`(`" and the end token is "`)`". We use the few-shot training samples from Wei et al. [4], which are general arithmetic word problems that are not from the MultiArith dataset. We evaluate on the full 600 samples provided in the dataset and compare to the Chain-of-Thought (CoT) and vanilla prompted results using the GPT-J model. Our To create the Self-Note prompts, we remove the Chain-of-Thought explanations and add our own Self-Note enrichments in the 8 Wei et al. [4] prompts. Table 12 shows the exact prompts used for the Self-Notes and Chain-of-Thought results. There is no validation set for the prompting experiments.

Table 8: Dataset Statistics. Note that the Math datasets (MultiArith and GSM8K) are used for few-shot prompting with GPT-J and GPT-3.

| Dataset | # train | # valid | # test | In domain | Out-of domain |
|---|---|---|---|---|---|
| Toy-story | 10k | 1k | 1k | 1, 2 hops | 3, 4 hops |
| Algorithm | 10k | 1k | 1k | 2-100 statements | 101-200 statements |
| Boolean Variable | 524k | 1k | 1k | 3-8 statements | 9-19 statements |
| Chess Piece | 200k | 1k | 1k | $\leq 80$ moves | $\geq 81$ moves |
| Chess Move | 200k | 1k | 1k | $\leq 80$ moves | $\geq 81$ moves |
| MultiArith | 8 (prompt) | N/A | 600 | N/A | N/A |
| GSM8K | 5 (prompt) | N/A | 1319 | N/A | N/A |

**GSM8K.** This task was introduced to evaluate more challenging arithmetic reasoning problems [19]. We selected 5 random samples in the training set to use as the few-shot demonstrations in the prompt. For Chain-of-Thought, we used the original solutions (minus the calculation annotations written inside "« »") given in the dataset. To create the Self-Note prompts, we remove the Chain-of-Thought explanations in the answer and add our own Self-Note enrichments in the 5 prompts. The start and end tokens are the same as MultiArith: " (" and ")". Table 13 shows the exact prompts used for the Self-Notes and Chain-of-Thought results. Self-Notes are only added after ".", "," and "?" tokens, so we do not have to call the API after each token.

## A.2 Training Details

For each non-prompting task, we train for a fixed 30 epochs with a learning rate of $2e$-5 and batch size of 32. For the Vanilla and Self-Notes methods, all models are trained on the context and answer tokens. For the Scratchpad method, the model is trained on the context and answer for Toy-Story, and is trained on only the answer for all other tasks since the scratchpad contains a full copy of the context. For the tasks that involve length generalization (Algorithmic, Boolean Variable, Chess Piecetype, Chess Move) we randomly offset the positional embedding IDs between 0 and 128 for the GPT-2 model during training. That is, the positional embedding ID of the first token is randomly assigned to a value from 0 to 128, and then continues sequentially. During inference, the position IDs always start at 0. This ensures that the model has seen all of the position embeddings that it will encounter during testing (following Kiyono et al. [38]). We note that this can also be solved by the relative position embeddings used in more recent models [49, 50].

We fine-tune all of the GPT-2 models on 8 NVIDIA V100 GPUs using an on-site cluster. The GSM8K experiments were done using the text-davinci-003 model with the OpenAI API, costing around $300.

*Reproducibility statement:* We will make code and data publicly available.

## A.3 Dummy Tokens: Separating Content and Compute

We introduced Self-Notes as a method for language models to do reasoning in the form of explicit tokens. To understand the value of these extra tokens, we seek to separate and measure the contribution of the additional compute that is allotted by these tokens from their content. In other words, we answer the question "Is it the specific tokens in the Self-Notes or is it simply extra steps of computation that is helping?". We do this by inserting the dummy token "_" at various locations throughout the context as an ablation. These are similar to a Self-Note with a single token that always remains the same.

We first consider the Toy-Story task. In the vanilla setting for this task, there are only facts and the question (e.g., "Bob is in the park. Bob has the key. Where is the key?"). The first comparison is inserting a dummy after every fact, which we call "Naïve Dummy" ("Bob is in the park. _ Bob has the key. _ Where is the key?"). The alternative, which we call "Self-Notes Dummy" is adding dummy tokens in the same locations where Self-Notes are written. In other words, at the locations where a relation between two facts can be inferred ("Bob is in the park. Bob has the key. _ Where is the key?"). Finally, we consider adding the dummy tokens where the Scratchpad tokens would be ("Bob is in the park. Bob has the key. Where is the key? _*"). This setting adds the same amount of dummy tokens as are used in Self-Notes Dummy (only 1 in the example described, but the "*" operator indicates that extra dummy tokens can be added). Figure 6 shows the results on the 3-hop and 4-hop test sets for the different dummy token settings compared to the natural

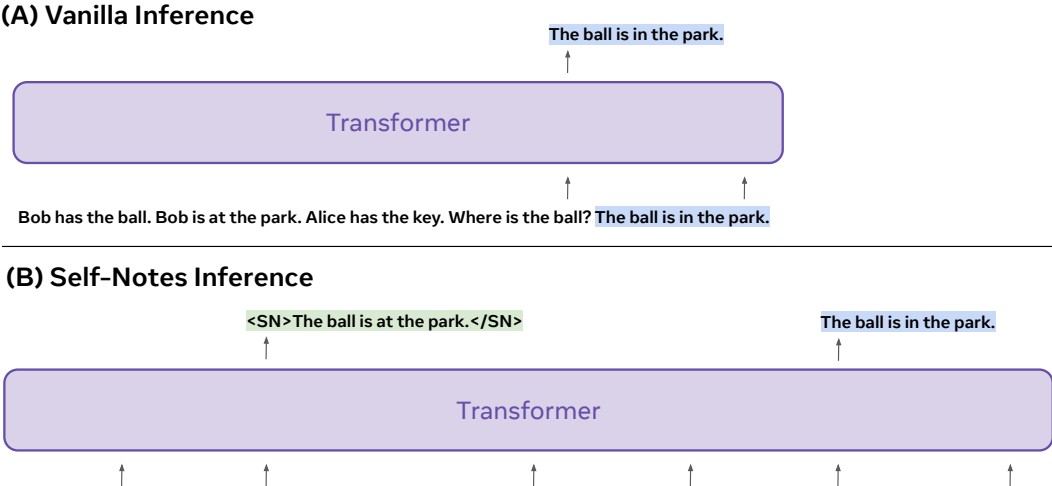

Figure 5: Comparison of Vanilla vs Self-Notes inference. Non-highlighted tokens are given to the model. Highlighted tokens are generated by the model. In Vanilla inference (top), the model generates only after the full context and question are provided. In Self-Notes inference (bottom), the model is able to generate after every word or sentence in the context. If the next most likely token is the Self-Notes start token "<SN>", then the model can autoregressively generate itself a note until the end token "</SN>" is generated, at which point it returns to reading the original context.

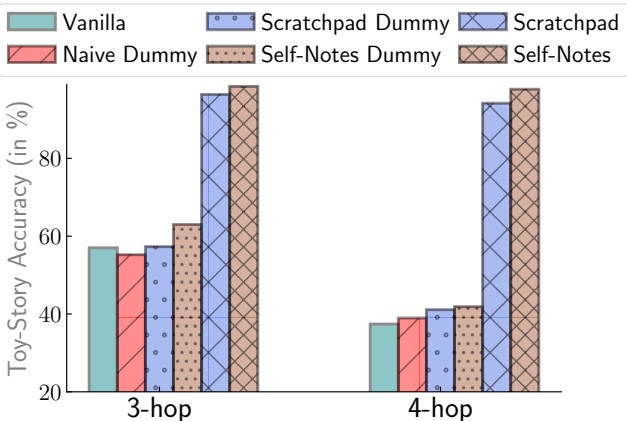

Figure 6: Ablation comparing the impact of (a) extra compute due to additional tokens, and (b) position of the additional tokens.

language tokens used by Self-Notes and Scratchpad. Intelligently inserting dummy tokens into the positions where Self-Notes should be performs the best out of the four settings. Importantly, it is better than inserting at the end of the context, where Scratchpad tokens are added. This alludes to the fact that *allowing the model to do extra computations in the middle of the context can be better than after the context*.[5] However, the gain from Self-Notes Dummy over other dummy variants pales in comparison to the gains of actual Self-Notes over its dummy counterpart, suggesting that the content of the intermediate notes matters more than just the additional compute.

We also test the usefulness of Self-Notes Dummy tokens in three other settings: Chess Piecetype, Chess Move, and WikiText-103 language modeling. For the chess experiments, we compare the

---

[5]In the in-context learning setting, Wei et al. [4] reported that gains with chain-of-thought were not due to additional compute. Our setting though differs in three aspects: (a) the additional compute is happening in the middle of the context in case of Self-Notes Dummy, (b) we're finetuning the language model, and (c) we're working with a much smaller language model.

vanilla moves ("`c2 c4`"), a dummy token between move positions ("`c2 _ c4`"), and the generated (non-dummy) piece tokens in Self-Notes ("`c2 P c4`"). For WikiText-103 language modeling, we compare the vanilla text ("`The cat sat on the mat`") with dummy tokens inserted between each word in the text ("`The _ cat _ sat _ on _ the _ mat`"). There are no ground-truth Self-Notes for WikiText-103, so we neglect such experiments. Table 10 shows the accuracy results for chess and perplexity for WikiText-103. Dummy tokens are not reported in the accuracy or perplexity numbers. For each task, dummy tokens improves the vanilla setting by a small margin.

## A.4  Further Ablations

**Oracle/No Self-Notes during inference.** We perform two ablations regarding the use of Self-Notes during inference. The first is an upper bound where we provide 100% Self-Notes supervision to the model during both training and inference (rather than just training). The second is where we give 100% Self-Notes supervision during training, but restrict the model from generating Self-Notes during inference. This baseline analyzes whether Self-Notes-augmented training data can still help the model learn about the task in its weights even without using Self-Notes at inference time. The results are shown in Table 11. As expected, oracle Self-Notes (100% Self-Notes supervision during inference) improves the performance. On the other hand, not allowing the model to generate Self-Notes leads to a drastic drop in performance due to distribution shift at inference time.

## A.5  Prompts for Few-Shot Self-Notes

For the results in Table 4 and Table 5, we use the Self-Notes prompts shown in Table 12 and Table 13, respectively. The MultiArith prompt (Table 12) is the same set of examples as used in Wei et al. [4]. We augment the standard prompts they use with Self-Notes, which are interleaved in the context and question. We use a GSM8K prompt (Table 13) derived from a subset of the training samples, where we added Self-Notes. All prompting experiments are compared to Vanilla (standard prompts) and Chain-of-thought (COT) prompts.

Table 9: Test sample from the Toy-story task. In this example, the question ($Q$) is "Where is the ball?" and the answer ($A$) is "the ball is at the farm.". The vanilla model fails at multi-step reasoning and incorrectly predicts that the ball is at the "store" The Scratchpad starts to incorrectly reason where the ball is early, and cannot correctly predict where it ends. The Self-Notes method writes each intermediate reasoning step correctly, including the correct location of the ball, before the final question is asked.

| Model | Context | Prediction |
|---|---|---|
| Vanilla (original context) | Mary is with Daniel. Frank is with Sandra. John has the book. Frank has the suitcase. Daniel is at the station. the banana is inside the basket. Bob has the apple. Bob has the bag. the ball is inside the box. the apple is inside the bag. Alice has the banana. Alice has the key. John is at the farm. Charlie is at the bridge. the book is inside the box. Alice is at the store. Bob is with Alice. | the ball is at the store. |
| Scratchpad (scratchpad context) | [SQ: Where is Mary? A: Mary is at the station. SQ: Who has the ball? A: Bob has the ball. SQ: Who has the basket? A: Alice has the basket. SQ: Where is the book? A: the book is at the farm. SQ: Where is the bag? A: the bag is at the ball. SQ: Where is the ball? A: the ball is at the bag. SQ: Who has the box? A: John has the box. SQ: Where is the box? A: the box is at the farm. SQ: Who has the key? A: Alice has the key. SQ: Where is the banana? A: the banana is at the store. SQ: Where is the basket? A: the basket is at the store. SQ: Where is the key? A: the key is at the store. SQ: Where is Bob? A: Bob is at the store. SQ: Where is the apple? A: the apple is at the store. SQ: Where is the suitcase? A: the suitcase is at the store. SQ: Where is the basket? A: the basket is at the store.] | the ball is at the ball. |
| Self-Notes | Mary is with Daniel. Frank is with Sandra. John has the book. Frank has the suitcase. Daniel is at the station. SQ: Where is Mary? Mary is at the station. the banana is inside the basket. Bob has the apple. Bob has the bag. the ball is inside the box. the apple is inside the bag. Alice has the banana. SQ: Who has the basket? Alice has the basket. Alice has the key. John is at the farm. SQ: Where is the book? the book is at the farm. Charlie is at the bridge. the book is inside the box. SQ: Who has the box? John has the box. SQ: Where is the box? the box is at the farm. SQ: Who has the ball? John has the ball. SQ: Where is the ball? the ball is at the farm. Alice is at the store. SQ: Where is the banana? the banana is at the store. SQ: Where is the basket? the basket is at the store. SQ: Where is the key? the key is at the store. Bob is with Alice. SQ: Where is Bob? Bob is at the store. SQ: Where is the apple? the apple is at the store. SQ: Where is the bag? the bag is at the store. SQ: Where is the key? the key is at the store. | the ball is at the farm. |

Table 10: Results with Dummy Tokens. For the chess tasks, we report the results for the OOD setting (over 80 moves).

| Task | Vanilla | Dummy | Self-Notes |
|---|---|---|---|
| **Chess Piecetype** | 82.9 ±2.3 | 84.8 ±1.7 | **94.8** ±0.7 |
| **Chess Move** | 39.8 ±0.2 | 40.4 ±1.2 | **41.8** ±0.9 |
| **WikiText-103** | 25.9ppl | **24.9ppl** | n/a |

Table 11: Ablation comparing the performance of Self-Note with (i) ground truth Self-Notes at test time, and (ii) abstaining from generation of Self-Notes during inference.

| Self-Notes % | | Algorithmic | | Toy-Story | |
|---|---|---|---|---|---|
| **Train** | **Test** | **2-100** | **101-200*** | **3-hop*** | **4-hop*** |
| 100% | 100% | 100.0 ±0.0 | 100.0 ±0.0 | 99.9 ±0.1 | 99.8 ±0.3 |
| 100% | none | 21.3 ±0.6 | 9.2 ±0.5 | 37.7 ±3.9 | 29.0 ±1.6 |

Table 12: MultiArith examples used for the few-shot prompting experiments. To get the Vanilla prompts, we simply remove the highlighted reasoning tokens. Not only does Self-Notes result in better performance in most cases, it can also require fewer generated reasoning tokens than CoT or Scratchpad since the tokens are already inline with the context, as demonstrated in this table.

| Chain-of-thought (CoT) | Self-Notes |
|---|---|
| Q: There are 15 trees in the grove. Grove workers will plant trees in the grove today. After they are done, there will be 21 trees. How many trees did the grove workers plant today?
A: There are 15 trees originally. Then there were 21 trees after some more were planted. So there must have been 21 - 15 = 6. The answer is 6. | Q: There are 15 trees in the grove. (15 total) Grove workers will plant trees in the grove today. After they are done, there will be 21 trees. (21 - 15 = 6 left) How many trees did the grove workers plant today?
A: The answer is 6. |
| Q: If there are 3 cars in the parking lot and 2 more cars arrive, how many cars are in the parking lot?
A: There are originally 3 cars. 2 more cars arrive. 3 + 2 = 5. The answer is 5. | Q: If there are 3 cars in the parking lot and 2 more cars arrive, (3 + 2 = 5 total) how many cars are in the parking lot?
A: The answer is 5. |
| Q: Leah had 32 chocolates and her sister had 42. If they ate 35, how many pieces do they have left in total?
A: Originally, Leah had 32 chocolates. Her sister had 42. So in total they had 32 + 42 = 74. After eating 35, they had 74 - 35 = 39. The answer is 39. | Q: Leah had 32 chocolates and her sister had 42. (32 + 42 = 74 total) If they ate 35, (74 - 35 = 39 left) how many pieces do they have left in total?
A: The answer is 39. |
| Q: Jason had 20 lollipops. He gave Denny some lollipops. Now Jason has 12 lollipops. How many lollipops did Jason give to Denny?
A: Jason started with 20 lollipops. Then he had 12 after giving some to Denny. So he gave Denny 20 - 12 = 8. The answer is 8. | Q: Jason had 20 lollipops. (20 total) He gave Denny some lollipops. Now Jason has 12 lollipops. (20 - 12 = 8 left) How many lollipops did Jason give to Denny?
A: The answer is 8. |
| Q: Shawn has five toys. For Christmas, he got two toys each from his mom and dad. How many toys does he have now?
A: Shawn started with 5 toys. If he got 2 toys each from his mom and dad, then that is 4 more toys. 5 + 4 = 9. The answer is 9. | Q: Shawn has five toys. (5 total) For Christmas, he got two toys each from his mom and dad. (2 + 2 = 4 more) How many toys does he have now? (5 + 4 = 9 total)
A: The answer is 9. |
| Q: There were nine computers in the server room. Five more computers were installed each day, from monday to thursday. How many computers are now in the server room?
A: There were originally 9 computers. For each of 4 days, 5 more computers were added. So 5 * 4 = 20 computers were added. 9 + 20 is 29. The answer is 29. | Q: There were nine computers in the server room. (9 total) Five more computers were installed each day, from monday to thursday. (5 * 4 = 20 more) How many computers are now in the server room? (9 + 20 = 29 total)
A: The answer is 29. |
| Q: Michael had 58 golf balls. On tuesday, he lost 23 golf balls. On wednesday, he lost 2 more. How many golf balls did he have at the end of wednesday?
A: Michael started with 58 golf balls. After losing 23 on tuesday, he had 58 - 23 = 35. After losing 2 more, he had 35 - 2 = 33 golf balls. The answer is 33. | Q: Michael had 58 golf balls. (58 total) On tuesday, he lost 23 golf balls. (58 - 23 = 35 left) On wednesday, he lost 2 more. (35 - 2 = 33 left) How many golf balls did he have at the end of wednesday?
A: The answer is 33. |
| Q: Olivia has $23. She bought five bagels for $3 each. How much money does she have left?
A: Olivia had 23 dollars. 5 bagels for 3 dollars each will be 5 x 3 = 15 dollars. So she has 23 - 15 dollars left. 23 - 15 is 8. The answer is 8. | Q: Olivia has $23. (23 total) She bought five bagels for $3 each. (5 * 3 = 15 total) How much money does she have left? (23 - 15 = 8 left)
A: The answer is 8. |

Table 13: GSM8K examples used for the few-shot prompting. Reasoning tokens are highlighted by green. To get the Vanilla prompts, we simply remove the highlighted reasoning tokens.

| Chain-of-thought (CoT) | Self-Notes |
|---|---|
| Q: A restaurant is counting their sales for the day. They sold 10 meals at $8 each, 5 meals at $10 each, and 20 meals at $4 each. In dollars, how much money did the restaurant make throughout the day?

A: From the first set of meals, the restaurant earned 10 meals * $8 = $80. From the second set of meals, the restaurant earned 5 meals * $10 = $50. From the third set of meals, the restaurant earned 20 meals * $4 = $80. In total, the restaurant earned $80 + $50 + $80 = $210. So the answer is 210 | Q: A restaurant is counting their sales for the day. They sold 10 meals at $8 each, (that's 10 meals * $8 = $80 total sale) 5 meals at $10 each, (that's 5 meals * $10 = $50 additional sale. So $80 + $50 = $130 total sale so far) and 20 meals at $4 each. (that's 20 meals * $4 = $80 more sale. So $130 + $80 = $210 total sale so far) In dollars, how much money did the restaurant make throughout the day? (the total sale so far is 210 dollars)

A: The answer is 210 |
| Q: Paul lives in a 5th story apartment. He makes 3 trips out from and back to his apartment throughout the day each day of a week. How many feet does he travel vertically in total over the week if each story is 10 feet tall?

A: Since Paul makes 3 trips per day, and each trip involves going both down and up, this means he travels the full vertical distance of his apartment complex 3 * 2 = 6 times a day. Since there are 7 days in a week, this means he makes this trip 6 * 7 = 42 times a week. Since each story is 10 feet tall, that means with 5 stories he travels 5 * 10= 50 feet each trip. Since he made 42 trips of 50 feet each, this means he traveled 50 * 42 = 2100 feet in a week. So the answer is 2100 | Q: Paul lives in a 5th story apartment. He makes 3 trips out from and back to his apartment throughout the day each day of a week. (he travels the full vertical distance of his apartment complex 3 * 2 = 6 times a day) How many feet does he travel vertically in total over the week if each story is 10 feet tall? (there are 7 days in a week, this means he makes this trip 6 * 7 = 42 times a week. Since each story is 10 feet tall, that means with 5 stories he travels 5 * 10 feet = 50 feet each trip. Since he made 42 trips of 50 feet each, this means he traveled 50 feet * 42 = 2100 feet)

A: The answer is 2100 |
| Q: The Tampa Bay Bucs have 13 football players and 16 cheerleaders. If 10 football players and 4 cheerleaders quit, how many football players and cheerleaders are left?

A: There are 13 - 10 = 3 football players left. There are 16 - 4 = 12 cheerleaders left. In total there are 3 + 12 = 15 football players and cheerleaders left. So the answer is 15 | Q: The Tampa Bay Bucs have 13 football players and 16 cheerleaders. If 10 football players and 4 cheerleaders quit, (there are 13 - 10 = 3 football players left. And 16 - 4 = 12 cheerleaders left) how many football players and cheerleaders are left? (in total 3 + 12 = 15 football players and cheerleaders left)

A: The answer is 15 |
| Q: Leila and her friends want to rent a car for their one-day trip that is 150 kilometers long each way. The first option for a car rental costs $50 a day, excluding gasoline. The second option costs $90 a day including gasoline. A liter of gasoline can cover 15 kilometers and costs $0.90 per liter. Their car rental will need to carry them to and from their destination. How much will they save if they will choose the first option rather than the second one?

A: Leila and her friends will travel a total distance of 150 x 2 = 300 kilometers back-and-forth. They will need 300 / 15 = 20 liters of gasoline for this trip. So, they will pay $0.90 x 20 = $18 for the gasoline. Thus, the first option will costs them $50 + $18 = $68. Therefore, they can save $90 - $68 = $22 if they choose the first option. So the answer is 22 | Q: Leila and her friends want to rent a car for their one-day trip that is 150 kilometers long each way. (the total travel distance is 150 x 2 = 300 kilometers) The first option for a car rental costs $50 a day, excluding gasoline. The second option costs $90 a day including gasoline. A liter of gasoline can cover 15 kilometers and costs $0.90 per liter. Their car rental will need to carry them to and from their destination. (they will need 300 / 15 = 20 liters of gasoline. So, they will pay $0.90 x 20 = $18 for the gasoline) How much will they save if they will choose the first option rather than the second one? (the first option will costs them $50 + $18 = $68 total. Therefore, they can save $90 - $68 = $22 if they choose the first option)

A: The answer is 22 |
| Q: Daniel has a collection of 346 video games. 80 of them, Daniel bought for $12 each. Of the rest, 50% were bought for $7. All others had a price of $3 each. How much did Daniel spend on all the games in his collection?

A: On 80 games, Daniel spend 80 games * $12/game = $960. The rest of the collection is 346 games - 80 games = 266 games. 50% of these games means 50 / 100 * 266 games = 133 games. Daniel bought them for $7 each, so he had to spend 133 games * $7/game = $931 on them. The other 133 games were bought for $3 each, so they've cost him 133 games * $3/game = $399. On all games in total Daniel spent $960 + $931 + $399 = $2290. So the answer is 2290 | Q: Daniel has a collection of 346 video games. 80 of them, Daniel bought for $12 each. (that means he spent 80 games * $12/game = $960 in total) Of the rest, (the rest is 346 games - 80 games = 266 games) 50% were bought for $7. (that means 50 / 100 * 266 games = 133 games bought for $7 each. So he had to spend 133 games * $7/game = $931. The total so far is $960 + $931 = $1891) All others had a price of $3 each. (there are 133 games remaining and each were bought for $3 each, so they've cost him 133 games * $3/game = $399. The total so far is $1891 + $399 = $2290) How much did Daniel spend on all the games in his collection? (he spent $2290 for games in total)

A: The answer is 2290 |

