# OpenReview forum: "Learning to Reason and Memorize with Self-Notes"
_NeurIPS.cc/2023/Conference — NeurIPS 2023 poster_

### Official Review · Reviewer_v6D1 · 2023-06-12

**Soundness:** 2 fair
**Presentation:** 4 excellent
**Contribution:** 2 fair
**Rating:** 5
**Confidence:** 4

**Summary:**

The paper presents a general prompting approach for LLMs: instead of generating intermediate "thoughts" after processing the prompt as Chain-of-Thought (CoT), this paper proposes adding "Self-Notes" **while reading the prompt**. That is, while the initial reading of the prompt, the model can generate intermediate Self-Notes that defer the reading of the rest of the prompt until the end of generating the Self-Note.

The approach is evaluated across multiple reasoning benchmarks, with supervised models, semi-supervised models, unsupervised models (which I'd call "unlabeled" models, because they *are* supervised on an augmented version of the supervised input), and few-shot prompted models.

**Strengths:**

## Strengths
* The idea to generate Self-Notes *while processing the input*, rather than at the end, is simple and novel. It is inspiring to see how such a simple idea has not been used before.
* The paper evaluates multiple datasets and tasks: Toy-Story (a synthetic multi-hop QA), algorithmic & boolean (a synthetic source code simulation), Chess Piecetype & Chess Move (a synthetic chess simulation).
* The paper evaluates multiple supervision scenarios: supervised models, semi-supervised models, unsupervised models (which I'd call "unlabeled" models, because they *are* supervised on an augmented version of the supervised input), and few-shot prompted models.
* The Related Work places the paper well in the related literature.
* The paper is well-written and easy to follow

**Weaknesses:**

## Weaknesses
* The main weakness is the relatively small models that were used:
    * In supervised and semi-supervised settings, the base model for both the baseline and the model that Self-Notes is applied on is a **GPT-2-base** (there is actually no `gpt2-base` model on Huggingface. Do the authors mean `https://huggingface.co/gpt2`? If so, this is the **smallest** version of GPT-2, with 124M parameters). While providing a good initial indication, I am not sure that the conclusions for GPT-2 will hold for larger models (because of "emergent abilities" and generally very different behavior of GPT-2-sized models compared to larger models). Even if the authors have access only to a single GPU, I think that there are larger models that can be fine-tuned on a single GPU.
    * The few-shot prompting experiments were performed on **GPT-J**, which is a bit disappointing - these days, it is relatively easy for the broad community to experiment with large models via prompting API, and new few-shot prompting approaches such as Self-Notes can be easily evaluated on much stronger models such as GPT-3/4 / PaLM / Claude / Codex.
    * For the GSM-8K benchmark, the few-shot prompting experiments were performed using **GPT-3** (since not mentioned otherwise, I am assuming `text-davinci-001`). Since these experiments could be easily evaluated using `text-davinci-002`, `text-davinci-003`, `code-davinci-002` by just replacing the model name, I am left with the conclusion is that Self-Notes does not work with these stronger models.

While these experiments are extensive, I believe that they do not confirm the strong claims of the paper, which hurts its soundness.

**Questions:**

## Questions

1. Wouldn't the approach be stronger if the question was provided **first**, before the context input? Currently, "The model can use notes as a form of working memory by **writing information that might be useful in the future**" (Lines 76-77). But how would the model know what information might be useful in the future, if it hadn't processed the question yet?
2. I would rename the "unsupervised Self-Notes" to "Unlabeled Self-Notes" or something in this spirit, because the model **is** supervised, but as far as I understand, on augmentation of the dataset, rather than on a manually-labeled version of the dataset.
3. Imagine that you had the ability to run GPT-3/4 locally. Isn't expecting the model to generate self-notes much slower than standard prompting? in standard prompting, the model reads the entire prompt in a single forward pass, where all prompt tokens can be processed in parallel. In Self-Notes, don't we must feed the prompt tokens one-by-one, to check if the model has predicted the next token to be "start-of-note"? (and in that case, stop feeding the original prompt, and feed the generated note instead?)

## Summary
Overall, the idea presented in this paper is really nice and novel. I also appreciate the authors' empirical efforts across various benchmarks and supervision scenarios.

Unfortunately, the idea seems not to work with models newer than the first version of GPT-3 and supervised models that are larger than `gpt2` (small), and thus I gave low "soundness" and "contribution" scores, and overall a "weak accept" score.

I would have given a higher score if the paper had demonstrated that the same idea can be useful to improve stronger models such as `text-davinci-002`, `text-davinci-003`, `code-davinci-002`, `gpt-3.5-turbo`, `gpt-4`, and supervised models that are larger than `gpt2` (even `gpt2-large` / `gpt2-xl` can be fine-tuned on a single GPU, I believe).

I hope that the authors would strengthen the evaluation for the next version by evaluating with stronger base models. Although I vote for acceptance, I will understand if the other reviewers will argue for rejection because of these evaluation limitations.

---

> ### Author Rebuttal · Authors · 2023-08-10
>
> Thank you for your comprehensive review. In particular, we appreciate the thorough and well understood list of our contributions including our empirical efforts across various benchmarks and supervision scenarios. We have addressed your comments regarding the GPT3+ and other OpenAI models in the response to all reviewers.
>
> ---
>
> > In supervised and semi-supervised settings, the base model for both the baseline and the model that Self-Notes is applied on is a GPT-2-base (there is actually no gpt2-base model on Huggingface. Do the authors mean https://huggingface.co/gpt2? If so, this is the smallest version of GPT-2, with 124M parameters). While providing a good initial indication, I am not sure that the conclusions for GPT-2 will hold for larger models (because of "emergent abilities" and generally very different behavior of GPT-2-sized models compared to larger models). Even if the authors have access only to a single GPU, I think that there are larger models that can be fine-tuned on a single GPU.
>
> Thank you for pointing this out. We do indeed use the smallest GPT-2 model (124M parameters), and will update our manuscript to reflect so. While it’s true in general that there are different properties in larger models, we observe that the GPT-2 result trends are consistent with the few-shot prompting experiments on the larger models. The main objective of this work is to experimentally validate the difference between in-context thoughts (Self-Notes) vs post-context thoughts (CoT/Scratchpad) on a fixed model, which we do across a variety of models of different sizes.
>
> ---
>
> > The few-shot prompting experiments were performed on GPT-J, which is a bit disappointing - these days, it is relatively easy for the broad community to experiment with large models via prompting API, and new few-shot prompting approaches such as Self-Notes can be easily evaluated on much stronger models such as GPT-3/4 / PaLM / Claude / Codex.
>
> We did evaluate using a GPT-3 model (text-davinci-001). Note that for cost as well as model openness reasons, we didn’t evaluate our models on a whole array of models accessible via a paid API.
>
> ---
>
> > Wouldn't the approach be stronger if the question was provided first, before the context input? Currently, "The model can use notes as a form of working memory by writing information that might be useful in the future" (Lines 76-77). But how would the model know what information might be useful in the future, if it hadn't processed the question yet?
>
> Thank you for raising this great question. We agree that a question-driven input, i.e. input placed first, should filter irrelevant Self-Notes for the entity(ies) of interest in the question. However, we’re imagining a curious reader which takes these general notes, and which can ultimately answer questions of a wider range. We note that this can be beneficial for several reasons. First, in many scenarios because the model has already precomputed the answer to a variety of questions, and doesn’t have to “re-reason” each time a new question is asked.
> Second, for certain tasks, such as program evaluation, it can be hard to predict what entities (variables) can influence the value of the entity of interest since that would require predicting the evolution of the program. Finally, by providing the question after the context, we can evaluate a fair comparison to previous CoT/Scratchpad methods which do the same.
>
> ---
>
> > I would rename the "unsupervised Self-Notes" to "Unlabeled Self-Notes" or something in this spirit, because the model is supervised, but as far as I understand, on augmentation of the dataset, rather than on a manually-labeled version of the dataset.
>
> Thank you for bringing this up. We see from your description that the term Unsupervised Self-Notes could be interpreted the wrong way. The model is trained with a supervised loss on the answer, but the self-notes are unsupervised. We will update this in our final manuscript.
>
> ---
>
> > Imagine that you had the ability to run GPT-3/4 locally. Isn't expecting the model to generate self-notes much slower than standard prompting? in standard prompting, the model reads the entire prompt in a single forward pass, where all prompt tokens can be processed in parallel. In Self-Notes, don't we must feed the prompt tokens one-by-one, to check if the model has predicted the next token to be "start-of-note"? (and in that case, stop feeding the original prompt, and feed the generated note instead?)
>
> Self-Notes is expected to run slower than standard prompting since it generates additional tokens, and as suggested, the input can’t be processed in parallel. Note that we only check for a Self-Note generation at the end of statements/sentences which makes it more efficient than performing this check after every token. Comparison with Scratchpad/CoT is task-dependent though. In our experiments, we found Self-Notes to be faster than scratchpad for certain tasks (e.g. Algorithmic where scratchpad has to copy the entire program), and slower for others (e.g. Toy Story).  Speedups can also be achieved by caching hidden states of previous forward passes.

---

> > ### Comment · Reviewer_v6D1 · 2023-08-10
> > **Discussion period**
> >
> > Dear authors,
> > Thank you for your response.
> >
> > I appreciate your efforts and I like the elegant and simple idea proposed in this paper, but I am left with the conclusion that Self-Notes does not work with stronger models than `GPT2-small` and `GPT-3`.
> >
> > >"we observe that the GPT-2 result trends are consistent with the few-shot prompting experiments on the larger models"
> >
> > I do not agree with this claim, since no experiments were presented to support it, and there is much evidence that it is quite the opposite (e.g. ["Emergent Abilities", Wei et al., 2022](https://arxiv.org/pdf/2206.07682.pdf)).
> >
> > >We did evaluate using a GPT-3 model (text-davinci-001). Note that for cost as well as model openness reasons, we didn’t evaluate our models on a whole array of models accessible via a paid API.
> >
> > While I am a strong supporter of openness as well, I cannot accept these arguments:
> > 1. According to OpenAI [[pricing #1]](https://openai.com/pricing#language-models) [[pricing #2]](https://platform.openai.com/docs/deprecations/), **`text-davinci-001` is not cheaper than `text-davinci-002`, `text-davinci-003`, `gpt-3.5-turbo`**. The authors could easily run experiments with these models for the same cost, with a change of a single string.
> > 2. `code-davinci-002` is **free** and still accessible.
> > 3. There are various open-source models, across a variety of sizes, that the authors could have experimented with such as LLama, LLama-2, Vicuna, Falcon, etc.
> > 4. Models such as `gpt2-large` / `gpt2-xl` can be fine-tuned on a single GPU, and the authors persist to use only `gpt2-small` with no convincing justification.
> >
> > The authors argue that "there is nothing inherently limiting about our proposed method as we scale the size of the model up", which is true except for the fact that they did not demonstrate it, and the returns may diminish with larger models.
> >
> > The authors' refusal to consider experiments with newer and larger models leads me to no conclusion other than that Self-Notes does not work with stronger models than `GPT2-small` and `GPT-3`.
> > I hope that the authors will convince me otherwise by the end of this discussion period, or acknowledged that Self-Notes's benefits are limited to smaller models.
> > In the meantime, I am reducing my score.

---

> > > ### Author Response · Authors · 2023-08-21
> > >
> > > Thank you for your response. We have addressed your concern in a comment to all reviewers: https://openreview.net/forum?id=ZFwNdsDCRL&noteId=o2bvfXVmWz

---

### Official Review · Reviewer_PMpA · 2023-07-05

**Soundness:** 2 fair
**Presentation:** 2 fair
**Contribution:** 3 good
**Rating:** 6
**Confidence:** 5

**Summary:**

The authors introduce a method called "Self-Notes" that allows the model to think and write down its thoughts during the reasoning process. Unlike other approaches, this method enables the model to deviate from the input context, integrate previous reasoning steps, and enhance its memory with useful information. The experiments show that the Self-Notes method outperforms chain-of-thought and scratchpad methods by interleaving the input text with the model's notes.

**Strengths:**

1. The authors propose a method called "Self-Notes," which involves jotting down thoughts during the reasoning process.
2. The experiments compare their method with different algorithms and demonstrate the advantages of their approach in supervised, semi-supervised, and unsupervised learning settings.

**Weaknesses:**

1. The "Supervised Self-Notes" experiments were conducted using GPT-2. However, it is recommended to use larger open-source models like LLaMA for these experiments.
2. It is recommended to conduct "Semi-supervised Self-Notes" experiments for all tasks shown in the paper.
3. It is preferable to use the same evaluation set for all three types of experiments. For instance, Tables 4 and 5 utilize math word problems for evaluation, which are not included in Tables 2 and 3.

**Questions:**

1. The performance of GPT-3 is significantly low in table 5. What could be the reason behind this? Have you considered using GPT-3.5 for the experiments instead?
2. Is it possible to generalize the "Self-Notes" data from one task to different tasks?

**Limitations:**

The authors have discussed the limitation and broad impact in the paper.

---

> ### Author Rebuttal · Authors · 2023-08-10
>
> Thank you for your detailed review. We have addressed your comments regarding LLaMa and GPT-3.5 in the comment to all reviewers.
>
>
>
> ----
>
>
> > It is recommended to conduct "Semi-supervised Self-Notes" experiments for all tasks shown in the paper.
>
> Across the two tasks we conducted semi-supervised experiments on (Toy-Story and Algorithmic), we observed a consistent trend of Self-Notes outperforming the vanilla model at less than 25% supervision. The objective of these experiments was to show that it’s possible to achieve good performance even when we don’t have every sample in the training set labeled with Self-Notes, which can be expensive. We expect this trend to remain across other tasks.
>
>
> ----
>
>
> > It is preferable to use the same evaluation set for all three types of experiments. For instance, Tables 4 and 5 utilize math word problems for evaluation, which are not included in Tables 2 and 3.
>
> We did not run the fine-tuning settings (supervised, semi-supervised, unsupervised) on the math word problem datasets because they are small datasets (MultiArith only has 600 samples), and thus is better fit for few-shot prompting where we can evaluate on all samples.
>
> ----
>
>
> > The performance of GPT-3 is significantly low in table 5. What could be the reason behind this? Have you considered using GPT-3.5 for the experiments instead?
>
> Thank you for raising this question. Greedy decoding, 5-shot prompting instead of 8-shot prompting used by Wei et al. 2022, and use of text-davinci-001 are the main reasons for the reported performance. Our emphasis was to compare Self-Notes relative with CoT, rather than their absolute performance. See also the comment to all reviewers for more detail on this.
>
> ----
>
>
> > Is it possible to generalize the "Self-Notes" data from one task to different tasks?
>
> This is a great question. We expect that if the model is trained on a diversity of tasks with Self-Notes supervision, the model can generalize to novel tasks, as has been the case with instruction-tuned models that generalize to novel instructions when trained with a rich diversity of instruction-based tasks. That would be very exciting to see in future work.

---

> > ### Comment · Program_Chairs · 2023-08-10
> > **Discussion period**
> >
> > Dear Authors,
> >
> > Thank you for your response.
> >
> > My primary concern regarding the first weakness mentioned is unresolved in the rebuttal. If you do not want to use GPT-3.5, I'd recommend considering  test your method with LLaMa and LLaMA-2.
> >
> > To this end, I've adjusted my score from **5** to **4**. If this concern can be addressed during the discussion period, I am willing to retain my initial score of **5**.

---

> > > ### Author Response · Authors · 2023-08-21
> > >
> > > Thank you for your response. We have addressed your concern in a comment to all reviewers: https://openreview.net/forum?id=ZFwNdsDCRL&noteId=o2bvfXVmWz

---

> > > > ### Comment · Reviewer_PMpA · 2023-08-21
> > > > **Response to rebuttal**
> > > >
> > > > Thank you for including the experiments. I have no concerns regarding your work now. By the way, for the final version, I recommend testing the complete set using all llama2 models.

---

### Official Review · Reviewer_D5Ka · 2023-07-06

**Soundness:** 4 excellent
**Presentation:** 4 excellent
**Contribution:** 3 good
**Rating:** 7
**Confidence:** 4

**Summary:**

One very general and beneficial method for improving the outputs of LMs is called *chain-of-thought* (CoT) reasoning, by which an LM is trained or prompted to first output its step-by-step reasoning before outputting the answer to a problem. This paper proposes a major extension of CoT wherein the LM is trained or prompted to insert its reasoning (as *Self-Notes*) at any relevant points in the token stream, rather than waiting until the end of the prompt, which can be quite long when the problem description is long and complex and contains example solutions. Self-Notes also serve as a form of in-context memory, summarizing key inferences made by the model along the way. The new method is evaluated against LMs with and without CoT, on a wide range of datasets, and employing four different learning paradigms:  Supervised, semi-supervised, unsupervised, and few-shot prompted. The experiments clearly demonstrate significant performance gains due to Self-Notes in most settings, particularly in generalizing to cases not seen in training.

**Strengths:**

This work is sound and compelling. The presentation is especially clear and easy to follow. The results are particularly interesting given the current importance of LLMs, and the field’s focus on finding ways to improve LLM reasoning even further.

**Weaknesses:**

Most of the experiments were performed using GPT-2, which seriously lags behind the current generation of LLMs in terms of output quality. A few experiments in this work involved GPT-3, but the field has changed dramatically since the introduction of GPT-3.5 and especially GPT-4, which is particularly good at leveraging few-shot prompts. This work’s contribution could be much greater if the experiments included GPT-4.

Section 4.4 mentions that GPT-3 was called through the public API. Given this constrained access, Self-Notes were intentionally limited to appear only at end-of-sentence positions. The same could be done when using GPT-4 through the public API. But the small margin of improvement over CoT shown in Table 5 (using few-shot prompting) raises the question of how effective Self-Notes would be when limited to end-of-sentence positions using GPT-4.

All of the examples from the datasets in this work share a certain underlying structure:  step-by-step revelation of partial information, in a way that benefits from progressive inference and ongoing memory to track the evolving state. Such tasks seem ideally suited to the insertion of Self-Notes into the token stream. By contrast, the benefit of Self-Notes would be more surprising (and valuable) in the broader set of reasoning tasks that don’t share this special structure. For instance, consider the problem from section 8.2 in [2], where the LLM is tasked to modify exactly one number in an algebraic equation in order to make the right hand side equal a particular value. A chain of reasoning is required here, but the facts in the problem statement are not arranged in an obvious way to provide convenient insertion points for Self-Notes. The significance of Self-Notes would be greater if their benefit were shown to extend to other classes of important problems.


**Questions:**

What are the numeric results shown in Table 4 & Table 5? Accuracies?

**Limitations:**

No concerns.

---

> ### Author Rebuttal · Authors · 2023-08-10
>
> Thank you for your comprehensive and valuable review. We appreciate the nice summary and the comment highlighting the importance of improving LM reasoning with Self-Notes. We have addressed your comments regarding GPT-3.5 and GPT-4 in the response to all reviewers.
>
>
>
> ----
>
>
> >All of the examples from the datasets in this work share a certain underlying structure: step-by-step revelation of partial information, in a way that benefits from progressive inference and ongoing memory to track the evolving state. Such tasks seem ideally suited to the insertion of Self-Notes into the token stream. By contrast, the benefit of Self-Notes would be more surprising (and valuable) in the broader set of reasoning tasks that don’t share this special structure. For instance, consider the problem from section 8.2 in [2], where the LLM is tasked to modify exactly one number in an algebraic equation in order to make the right hand side equal a particular value. A chain of reasoning is required here, but the facts in the problem statement are not arranged in an obvious way to provide convenient insertion points for Self-Notes. The significance of Self-Notes would be greater if their benefit were shown to extend to other classes of important problems.
>
> We appreciate the careful analysis of the datasets we used. Self-Notes is a *general* method to allow language models to write itself notes at any time, as described in Figure 1 and nicely summarized in your review. It is easily generalizable to other tasks not described in this paper. In particular, we consider Chain-of-Thought to be a special case of Self-Notes, where the note can only come at the end of the context. For the algebraic problem described, the Self-Note could be written after it has seen the full context, which is similar to CoT. While the goal of this work was to compare to the types of tasks that CoT/Scratchpad are typically used for, implementing Self-Notes on other tasks such as the algebraic problem is an exciting direction and a great recommendation.
>
>
> ----
>
>
> > What are the numeric results shown in Table 4 & Table 5? Accuracies?
>
> Yes, these are accuracy (%), thank you for bringing this to our attention. We have updated the manuscript to reflect this.

---

> > ### Comment · Reviewer_D5Ka · 2023-08-18
> > **Response to rebuttal**
> >
> > Thank you for the clarifications!

---

### Official Review · Reviewer_7QVa · 2023-07-06

**Soundness:** 4 excellent
**Presentation:** 4 excellent
**Contribution:** 4 excellent
**Rating:** 8
**Confidence:** 4

**Summary:**

This paper introduces a variation to the chain of thought and scratchpad techniques that can easily be applied to pre-trained transformers.  While reading a passage, the model can insert "self notes" at any point in the input sequence.  These self-notes allow the model to perform chain-of-thought reasoning, by annotating the input sequence with additional information about characters, variables, etc.  Self-notes differ from standard COT because the model can create multiple notes, and place those notes at appropriate locations in the input sequence.

The authors compare several different ways of training a model to insert notes: supervised, semi-supervised, unsupervised, and prompting.  The "unsupervised" method is the most interesting, because it requires the least amount of human-generated annotations, although it does require that the model be fine-tuned on a QA dataset.  IMO, it's not quite "unsupervised", simply a clever way of using existing QA datasets to train a model to "ask itself questions".

All three methods produce improvements over SOTA on a variety of tasks.


**Strengths:**

This paper was very clearly written.  I especially applaud the authors for concisely illustrating their main contributions in a figure that appears right before the opening introductory paragraph.

The self-note mechanism is a fairly simple extension to prior work on scratchpad and chain-of-thought reasoning, but I fully expect it to be high-impact.  The technique is easy to understand, easy to implement, works with pre-trained models, and results in large and obvious improvements on reasoning tasks.  The experiments are well-designed; the authors compare multiple training methods on multiple tasks, and the chosen tasks are appropriate.


**Weaknesses:**

The main limitation of this work is (1) the requirement for human annotations in the supervised and semi-supervised cases, or (2) the requirement for a Q/A dataset in the unsupervised case.  The size of the models are relatively small (GPT2), and the tasks are fairly simple, and partially synthetic.  Additional work may be needed to scale this technique up, especially since existing QA datasets are quite small compared to the training corpus for modern LLMs.  However, I feel that some combination of unsupervised training and prompting could probably overcome the scaling challenges.


**Questions:**

Have you considered how this technique could be scaled to LLM size?  What potential problems do you expect to encounter?


**Limitations:**

The authors do not discuss societal impacts.  They do briefly mention scaling as an important area for future work, but do not go into any detail about the limitations of their training techniques.

---

> ### Author Rebuttal · Authors · 2023-08-10
>
> Thank you for the careful review and helpful comments. We appreciate the attention to detail, the concise description of contributions, and beneficial suggestions. Thank you also for highlighting the potential high-impact of our work on future research. We have addressed your comments regarding the GPT2 experiments and scaling to LLM size in the response to all reviewers.
>
>
>
> ----
>
>
>
> >The main limitation of this work is (1) the requirement for human annotations in the supervised and semi-supervised cases, or (2) the requirement for a Q/A dataset in the unsupervised case.
>
> Thank you for pointing out this important detail. We note that scratchpad and few-shot chain-of-thought also require human annotations and Q/A datasets. So while it is a drawback compared to vanilla training, it requires the same amount of annotations as Chain-of-Thought/Scratchpad. The goal of this work is to experimentally validate the difference between in-context thoughts (Self-Notes) vs purely post-context thoughts (CoT/Scratchpad). We have added this discussion to the manuscript.
>
>
>
> ----
>
>
>
> > The tasks are fairly simple, and partially synthetic.
>
> We aimed for a wide variety of tasks in several domains, demonstrating the advantage of our method in both synthetic and real-world tasks. We followed Fan et al. 2020 and Anil et al. 2022, for synthetic testbed tasks, and followed Wei et al. 2022 and Toshniwal et al. 2021 for real-world tasks. We specifically chose these datasets as we believe they most appropriately evaluate/demonstrate the ability of a model to do multi-step reasoning and state-tracking.
>
>
>
> ----
>
>
>
> > Additional work may be needed to scale this technique up, especially since existing QA datasets are quite small compared to the training corpus for modern LLMs. However, I feel that some combination of unsupervised training and prompting could probably overcome the scaling challenges.
>
> Thank you for raising this important point. There are several recent works showing that small-scale fine-tuning datasets (e.g. “LIMA”, “AlpaGasus”) work well with large pre-trained models. So we don’t expect there to be an issue fine-tuning large models on a small set of labeled Self-Note samples. We also show that we can get good performance with Self-Notes using few-shot prompting, where we only require a few labeled instances. We do agree that some combination of unsupervised training and prompting could lead to even greater gains, this is a great suggestion and an interesting direction to pursue.
>
>
>
> ----
>
>
>
> > The authors do not discuss societal impacts. They do briefly mention scaling as an important area for future work, but do not go into any detail about the limitations of their training techniques.
>
> Thank you for bringing this important missing discussion to our attention. We will add the following. We don’t expect any negative societal impacts as a direct result of our method. We acknowledge that there are already significant potential impacts with language models in general, but we don’t expect our work to mitigate any of such existing issues, other than perhaps adding an extra degree of interpretability by examining the self-notes themselves. As with all tokens generated by generative language models, there is a risk of hallucinations in the Self-Notes. The practical limitations of Self-Notes are similar to those of Chain-of-thought/Scratchpad. Namely, most applications require labeled Self-Note annotations in order for the model to learn how to generate them. Lastly, generating Self-Notes is more costly than vanilla Transformer inference, but can be more efficient than CoT/Scratchpad depending on the task.

---

> > ### Comment · Reviewer_7QVa · 2023-08-14
> >
> >
> > Thank you for these responses.

---

### Author Rebuttal · Authors · 2023-08-10

We would like to thank all reviewers for the invaluable feedback. We greatly appreciate the encouraging and helpful comments which have made our paper stronger. All reviewers pointed out the main contributions of our work and noted the potential impact of our work on future research. We respond to individual comments below each review, and make a general response to comments regarding our experimental setup here.

$\newline$

We show extensive and diverse experiments across 4 training and inference settings (supervised, semi-supervised, unsupervised, few-shot prompted), on 7 total datasets, using various models ranging from 124M to 175B parameters in size. Importantly, our proposed method applies generally across all model sizes, datasets, and training/inference settings that we have tested.

The fine-tuning experiments were performed with smaller models (GPT-2) since we use a large number of datasets and training settings, which takes a substantial amount of resources. The few-shot experiments were performed with larger models (GPTJ and GPT3 text-davinci-001), where a limiting factor was the financial constraints of a paid API.

There is an extensive amount of work which show that techniques to improve reasoning and state tracking improve as models are scaled up (Nye et al. 2021, Wei et al. 2022, Anil et al., 2022, Wang et al. 2023, Yao et al. 2023, Touvron et al. 2023). These works include both fine-tuning and few-shot prompting. Aside from being more expensive than vanilla training/inference, there is nothing inherently limiting about our proposed method as we scale the size of the model up.

We note that the GPT-4 API general availability wasn’t until July 6, 2023, so it was not considered for this submission. GPT-4 was also trained on GSM8K samples which include CoT explanations (OpenAI, 2023), and there is anecdotal evidence that this is the same for GPT-3.5 (Fu et al., 2023). We used a diverse set of models for our experiments to demonstrate the usefulness of our method. While there are more proprietary models (GPT3+, Claude, etc), they are not open, so we do not know the training data which may add unknown bias to results (e.g., techniques in Lightman et al., 2023), and are expensive so we can only run a certain amount of experiments, which we also unfortunately lacked funds for. Thus, in general we believe the community should push for the use of open models for reproducible science rather than necessitating the use of such models.

While models will always be improving and there will be new models to test a new method on, we ran extensive experiments and compared results for self-notes vs scratchpad/CoT for a fixed model. It seems clear that as models get stronger, they will be able to write better Self-Notes. We are hence excited about the applications of Self-Notes that will be enabled in the future.

---

> ### Comment · Reviewer_D5Ka · 2023-08-21
> **GPT-3.5 and GPT-4**
>
> Thank you for providing this explanation regarding other models. Since many readers will wonder why the work does not include GPT-3.5 and GPT-4, could you add a comment to the paper explaining the issues involved, perhaps in the Conclusion?

---

> > ### Author Response · Authors · 2023-08-21
> >
> > We would be happy to add this, thank you for the recommendation! While we have now additionally presented Llama2-70B results ([described above](https://openreview.net/forum?id=ZFwNdsDCRL&noteId=o2bvfXVmWz)), as we said there will always be other models to evaluate, and no paper can be exhaustive – or even close. As also said previously, in general we believe priority should be given to reproducible science and open methods where possible, whilst also being cognizant of privileged regimes when conducting large experiments – and that small model research (making small models better) is hence very important too!

---

### Author Response · Authors · 2023-08-21
**Author Rebuttal by Authors: Llama2-70B Results**

Thank you for the responses. We conducted additional few-shot prompting experiments using the very recent state-of-the-art open source model, Llama2-70B (Touvron et al., 2023), as per the request of the reviewers. For these additional experiments, we used the Algorithmic task, MultiArith, and GSM8K. For GSM8K and MultiArith, we use the 8-shot arithmetic reasoning prompt from Wei et al., 2023. The CoT and Vanilla prompts are the same as from Wei et al., and the Self-Notes prompt adapts the CoT prompt by putting some of the reasoning steps inside the context. For Algorithmic, we use a 5-shot prompt of randomly selected examples from our training set, ranging from 6-33 statements. We test on the 2-100 statement test set from our paper. In summary, we find that Self-Notes outperforms Chain-of-Thought (CoT) and the Vanilla baseline on all 3 tasks. The detailed Llama2-70B results are shown in the table below. We report accuracy (%) for each method.

| Dataset | Vanilla | CoT | Self-Notes |
| --------- | :----: | :----: | :----: |
| Algorithmic | 23.3 | 37.4 | __40.3__ |
| MultiArith | 33.3 | 94.5 | __96.2__ |
| GSM8K | 16.5 | 55.9 | __59.9__ |

---

> ### Comment · Reviewer_v6D1 · 2023-08-21
> **Great**
>
> Thank you for these additional experiments, I think they are very important.
>
> I increased my score.
>
> In the next version of the paper, I recommend extending the Llama-2 experiments to all experimental settings, and evaluating with GPT 3.5/4 as well, even though their training set is unknown.

---

> ### Comment · Reviewer_7QVa · 2023-08-21
> **Thank you.**
>
> Especially given the concerns of other reviewers, I also appreciate having these additional experimental results!

---

### Decision · Program_Chairs · 2023-09-21

**Decision:**

Accept (poster)

**Comment:**

This work receives mostly positive reviews and rebuttal has addressed some critical concerns. All reviewers agree that the idea self-notes is a nice extension to CoT that allows LLMs to take in-context notes during reasoning, and its effectiveness is evaluated on relatively small scale models. This work could be greatly improved with a systematic evaluation on latest LLMs, which should be included in the camera version as suggested by most reviewers. AC acknowledges the novelty and potential of this work, and thus recommend acceptance.